# An Efficient 3D Convolutional Neural Network for Dose Prediction in Cancer Radiotherapy from CT Images

**DOI:** 10.3390/diagnostics15020177

**Published:** 2025-01-14

**Authors:** Lam Thanh Hien, Pham Trung Hieu, Do Nang Toan

**Affiliations:** 1Faculty of Information Technology, Lac Hong University, Huynh Van Nghe, Bien Hoa 76120, Vietnam; lthien@lhu.edu.vn; 2Institute of Information Technology, Vietnam Academy of Science and Technology, Hoang Quoc Viet, Hanoi 10072, Vietnam; pthieu@ioit.ac.vn

**Keywords:** 3D deep learning model, CT images, dose prediction, U-Net architecture, residual connection, dose–volume histogram

## Abstract

**Introduction**: Cancer is a highly lethal disease with a significantly high mortality rate. One of the most commonly used methods for treatment is radiation therapy. However, cancer treatment using radiotherapy is a time-consuming process that requires significant manual work from planners and doctors. In radiation therapy treatment planning, determining the dose distribution for each of the regions of the patient’s body is one of the most difficult and important tasks. Nowadays, artificial intelligence has shown promising results in improving the quality of disease treatment, particularly in cancer radiation therapy. **Objectives**: The main objective of this study is to build a high-performance deep learning model for predicting radiation therapy doses for cancer and to develop software to easily manipulate and use this model. **Materials and Methods**: In this paper, we propose a custom 3D convolutional neural network model with a U-Net-based architecture to automatically predict radiation doses during cancer radiation therapy from CT images. To ensure that the predicted doses do not have negative values, which are not valid for radiation doses, a rectified linear unit (ReLU) function is applied to the output to convert negative values to zero. Additionally, a proposed loss function based on a dose–volume histogram is used to train the model, ensuring that the predicted dose concentrations are highly meaningful in terms of radiation therapy. The model is developed using the OpenKBP challenge dataset, which consists of 200, 100, and 40 head and neck cancer patients for training, testing, and validation, respectively. Before the training phase, preprocessing and augmentation techniques, such as standardization, translation, and flipping, are applied to the training set. During the training phase, a cosine annealing scheduler is applied to update the learning rate. **Results and Conclusions**: Our model achieved strong performance, with a good DVH score (1.444 Gy) on the test dataset, compared to previous studies and state-of-the-art models. In addition, we developed software to display the dose maps predicted by the proposed model for each 2D slice in order to facilitate usage and observation. These results may help doctors in treating cancer with radiation therapy in terms of both time and effectiveness.

## 1. Introduction

Nowadays, cancer is one of the most dangerous diseases and a concern not only for economically challenged countries but also worldwide. It is the leading cause of death worldwide: according to a report [1], cancer accounted for nearly 10 million deaths in 2020, and approximately one-third of cancer-related deaths were attributed to tobacco use, alcohol consumption, and unhealthy lifestyles. Although 2020 was the year of the COVID-19 pandemic [2], cancer remained a silent killer, responsible for a significant number of deaths. Nevertheless, numerous cancer cases that were identified and treated in their early stages have been successfully cured [3]. In terms of definition, cancer is a disease in which some of the body’s abnormal cells grow uncontrollably and form a mass or tumor. Cancer can start almost anywhere in the human body, which consists of trillions of cells [4,5]. These cells gradually destroy and invade healthy tissues within the body, spreading from nearby organs to distant parts of the body (metastasis) [6].

Currently, there are numerous cancer treatment methods available, including chemotherapy, radiation therapy, conventional surgery, immunotherapy using drugs, and more [7]. Among these, radiation therapy is a common and widely used approach [8]. Radiation therapy is a medical treatment that involves the use of high-energy radiation to target and damage cancerous cells, thereby inhibiting their ability to grow and divide [9]. It has consistently proven to be a powerful and reliable treatment for cancer, providing significant therapeutic benefits for over a hundred years, and it continues to play a vital role in cancer care today [10]. This treatment is carried out by specialized radiation machines [11]. In contrast to other cancer treatments that affect the entire body, radiation therapy is typically a localized treatment [12]. This means that it primarily targets and affects only the specific area of the body where the tumor is located, minimizing the impact on surrounding healthy tissues. Radiation therapy is a process that consumes a significant amount of time, not only in the planning stages but also throughout the treatment process [13,14]. It can take many days to calculate and determine the distribution of the radiation dose that meets the optimal clinical standards. Furthermore, its accuracy heavily depends on the experience and skills of the treatment planners and doctors [15]. Estimating the radiation dose is an extremely crucial process, where treatment planners must determine the necessary radiation dose for the target regions/tumors while minimizing the radiation dose to healthy surrounding cells/organs.

Nowadays, artificial intelligence has been incorporated into research and applications in many tasks in cancer radiation treatment, especially for predicting the necessary dose distribution. In 2019, Jiang Jue et al. [16] developed a novel block-wise self-attention approach and applied it in a U-Net model [17] to segment normal organ structures from head and neck CT scans. In the paper, the authors showed that their method was computationally more efficient than many other methods. In 2023, Junkang Qin et al. [18] proposed a network model, CI-U-Net, that improves the accuracy of normal tissue segmentation. The authors used the tomographic abdominal organ dataset Chaos [19], which is focused on abdominal organs such as the liver, kidney, and spleen structures. Also in 2023, Jie Liu et al. [20] presented the CLIP-Driven Universal Model for segmenting abdominal organs and detecting tumors. The model was created using a combination of 14 datasets, comprising a total of 3410 CT scans for training. It was subsequently evaluated on 6162 external CT scans. These datasets included images of 25 different organs and six types of tumors. In 2021, a deep learning method [21] for dose prediction was developed and demonstrated to accurately predict patient-specific doses for left-sided breast cancer. In the paper, the authors showed that the doses predicted by deep learning were superior to the results of the RapidPlan-generated VMAT plan. Another study focusing on dose prediction using Res-U-Net [22] was conducted by Toan DN and colleagues. The authors developed various data preprocessing and augmentation strategies to create an autonomous dose prediction system with their convolutional neural network.

This study addresses the challenge of predicting radiation doses from CT images in cancer treatment with radiation therapy. We approach the problem through deep learning, using an experimental dataset that includes head and neck cancer patients from the Open Knowledge-Based Planning Challenge. A custom 3D convolutional neural network model is proposed to automatically predict the radiation dose from a given CT image. To mitigate the vanishing gradient phenomenon during the training phase [23], we apply a mechanism called a residual connection [24] at certain locations in the model. The mean squared error is a common loss function for regression problems in deep learning. However, to ensure that the predicted dose has high significance in the context of radiation therapy, we use a new loss function based on a dose–volume histogram to train the model. Moreover, the experimental data were collected from a variety of hospitals, so it is necessary to perform preprocessing steps on the data before training. In summary, our contributions are as follows:We propose a custom 3D convolutional neural network model to automatically predict radiation doses from CT images in radiation therapy for cancer;We propose a loss function based on a dose–volume histogram to train the model;We evaluate the proposed method and compare it with several previous studies;We build software to visualize the dose map predicted by the model for easy viewing and use.

This paper is structured as follows. Section 2 presents the data preparation. Section 3 describes the methodology. Section 4 presents the experiment. Section 5 analyzes the obtained results. Section 6 presents the conclusions.

## 2. Data Preparation

The dataset used comprises 340 head and neck cancer patients who were treated with radiotherapy. It was sourced from TCIA [25], an open-access database containing medical imaging data for cancer research, managed by the University of Arkansas in the United States. The OpenKBP competition [26] cleaned and standardized the data, including the file structures and names, to create a consistent dataset for researchers to use and compare results. The dataset is divided into 200 samples for training, 100 for testing, and 40 for validation.

Common radiation therapy techniques used to treat head and neck cancer patients include CRT and IMRT. CRT uses imaging (such as CT or MRI scans) to create a detailed 3D map of the tumor and the surrounding normal tissues. The radiation is then shaped to match the contour of the tumor. This technique has the ability to target the tumor precisely, helping to limit the radiation dose to healthy tissues and protect critical structures such as the salivary glands, spinal cord, and eyes. IMRT is an advanced form of 3D conformal radiation therapy that uses varying intensities of radiation beams to target different parts of the tumor. Multiple beams are directed from different angles, and their intensities are adjusted based on the 3D model of the tumor. These radiation therapy techniques all require pre-determining the location of the tumor as well as the surrounding healthy organs.

Regarding the details of the dataset, each patient entry includes a CT image, healthy organs at risk (OARs), planning target volumes (PTVs), and an image showing the radiation therapy dose distribution corresponding to the CT image—this serves as the label of the dataset. The three-dimensional CT image has a size of 128 × 128 × 128, with voxel values ranging from 0 to 4095, where voxels with a value of 0 represent non-body locations. OARs are healthy organs located near tumors that are at high risk of damage during radiation, including seven organs: brain stem, spinal cord, right parotid gland, left parotid gland, larynx, esophagus, and mandible. PTVs are the target areas that radiation rays need to hit. Depending on the severity, the target area is divided into three areas: 70 Gy, 63 Gy, and 56 Gy. Each OAR and PTV is represented by a binary mask with a CT image size of 128 × 128 × 128, where a position with a value of 1 represents the location of the corresponding OAR or PTV and a value of 0 does not. Additionally, some patients may lack information about a certain OAR or PTV, whose corresponding binary mask will be all 0. The distribution of the radiotherapy dose is also represented by a 128 × 128 × 128 matrix, where each location represents the radiotherapy dose required for the corresponding location in the CT image, with dose values ranging from 0 to 70 Gy. Figure 1 illustrates a 2D slice image of a patient.

In order to increase the performance and learning ability of the model on the dataset, the following preprocessing and data augmentation techniques are applied:Data preprocessing is an important step in any deep learning problem. Real-world data are often incomplete and inconsistent due to many objective factors. Processing data before feeding them into a model can reduce the model’s training time and increase its inference capabilities on that dataset. There are many image preprocessing methods, so depending on the available data and the problem being considered, it is necessary to choose the appropriate method. In this study, the method used was standardization, a method used to change image intensity values. The CT images included in the dataset were taken by different scanners at different hospitals; a patient imaged with two different machines can produce completely different results due to differences in configuration and hardware factors between the tools. Standardization brings CT images to a common scale, through which the model can work more effectively on the dataset [22]. The standardization formula is $z=\frac{x-\mu }{\sigma }$; the voxel values after standardization have a mean of 0 and a standard deviation of 1.Data augmentation is the creation of additional data from existing data. Data scarcity is one of the common challenges in building deep learning models; too little data prevents the models from learning the generality of the problem (which can lead to overfitting [27]). The main causes of data scarcity are that the collection process for some specific types of data is too expensive or takes a lot of time or because such data rarely appear. There are many methods used to generate additional data, but for this problem of predicting the radiotherapy dose, methods that do not change the size and scale of the original image are preferred. Scaling methods introduce noise to the model and contribute nothing to the training process other than increasing complexity and runtime. The two methods we applied were image flipping and image translation, and the training dataset increased from 200 to 800.

## 3. Methods

### 3.1. Proposed Model

#### 3.1.1. The Custom 3D Convolutional Neural Network

Our proposed model includes two 3D variants of the U-Net architecture, which follow a typical encoder–decoder architecture. The first model (A) is used to predict the coarse dose, and the second model (B) is used to refine the output of Model A to obtain the final dose distribution. The input of Model A is a 128 × 128 × 128 three-dimensional image consisting of 11 channels, which are one CT image, seven binary masks of OARs, and three binary masks of PTVs.

The input (11 channels) and output (16 channels) of Model A are fed into Model B, so Model B has an input of 27 channels. Both Model A and Model B predict individual dose distributions, but only the output of Model B serves as the final result. The output of Model A is used in the optimization process and to reduce the computational burden on the network architecture during training. In detail, both Models A and B are U-Net networks consisting of a down-sampling path, an up-sampling path, and skip concatenations at the corresponding locations between the two paths to preserve information during inference and training. The architecture of our model can be observed in Figure 2.

The down-sampling paths of both Models A and B consist of five levels of feature maps. The input image undergoes feature extraction, and feature maps with sizes of 128 × 128 × 128, 64 × 64 × 64, 32 × 32 × 32, 16 × 16 × 16, and 8 × 8 × 8 are obtained. The residual connection is applied at the first three levels. Each of these levels consists of two 3-dimensional convolutions [28], with a kernel size of 3 × 3 × 3 and stride of 1, both of which preserve the size of the feature map. Each convolution is followed by an instance normalization (IN) layer [29] and a rectified linear unit (ReLU) activation function [30] (called a regular convolution block). With the residual connection mechanism, the feature map at the beginning of each level is connected to the feature map at the final position along the channel dimension to create a residual block. This connection allows the network to retain key information throughout the layers. As part of this process, the output of each residual block is reduced in size by half with max-pooling 2 × 2 × 2 [31]. By incorporating this residual connection mechanism, the network retains valuable information throughout the down-sampling path, which is crucial for the inference process. This approach helps mitigate the risk of losing important details as the data move through successive layers, ultimately improving the model’s performance in capturing and utilizing relevant features. The feature maps are obtained at levels four and five by applying max-pooling and convolution with a stride of 2, respectively. For Model A, the feature maps obtained at the beginning of each level have the following numbers of channels: 16, 48, 112, 240, and 496. For Model B, the corresponding numbers of channels are 32, 96, 224, 480, and 992.

The up-sampling path is used to gradually increase the resolution of the feature map obtained from down-sampling and ultimately produce the predicted dose image. Therefore, the feature maps in the up-sampling path also have five levels with sizes inverse to those in the down-sampling path: 8 × 8 × 8, 16 × 16 × 16, 32 × 32 × 32, 64 × 64 × 64, and 128 × 128 × 128. The resolution increase at each level is performed by linear interpolation [32] and is followed by a 3 × 3 × 3 convolution (without changing the feature map size), an instance normalization layer, and a ReLU function (called the up-sampling block). After the up-sampling block, there are two regular convolution blocks. To obtain more detailed information, feature maps with dimensions of 128 × 128 × 128, 64 × 64 × 64, 32 × 32 × 32, and 16 × 16 × 16 in the down-sampling path are passed to the up-sampling path at the corresponding locations via skip concatenations. Feature maps generated by the encoder are crucial for refining the spatial information of the input data. To ensure that important details are preserved, skip connections are used to directly transfer these feature maps from the encoder to the decoder. This process helps counteract the loss of spatial information that typically occurs during the down-sampling phase of the encoder. By supplementing the decoder with feature maps from the encoder, these skip concatenations allow the model to recover finer details and improve the accuracy of location-based predictions. In the up-sampling path, for Model A, each feature map has corresponding channel numbers of 496, 240 + 128, 112 + 64, 48 + 32, and 16 + 16, and Model B has corresponding channel numbers of 480 + 256, 224 + 128, 96 + 64, and 32 + 32. The predicted dose image is generated by applying a 1 × 1 × 1 convolutional layer to the outputs of Models A and B, respectively. The number of channels of feature maps in Model A is only half that of Model B because Model A is designed to generate coarse predictions. In the up-sampling path, the residual connection mechanism is not applied because the skip concatenations between the down-sampling and up-sampling paths already store and preserve information. In addition, to ensure that the predicted dose does not have negative values, a ReLU function is applied to the output of both Models A and B to turn negative values into zero.

#### 3.1.2. Cascade Learning

Cascade learning [33] allows us to train deep networks more quickly and achieve convergence more efficiently. The idea behind this algorithm is that it divides the network into parts and trains each part sequentially until the weight updates for those parts become negligible, or the metrics on the validation set stabilize. This training strategy helps deal with the vanishing gradient problem by forcing the model to learn data features at each part of the network architecture. An overview of cascade learning can be observed in Figure 3. The learning process takes place by taking the first layer of the model, connecting it to an output, and then training it until the weights of that layer converge. Next, the second layer (starting from the input image) is connected to an output and trained. This process is repeated until all layers converge.

For our custom convolutional neural network, we split the full network architecture into two parts, which are, respectively, 2 U-Net-based models, with an additional output for the first model (Model A). Furthermore, for the second model (Model B), we not only use the feature map obtained from Model A but also include the input image to provide additional information during the training process. The training process is carried out simultaneously for both models, and each model uses a separate loss function. More details are provided in Section 3.2.

#### 3.1.3. Residual Connection

In general, U-Net architectures contain many layers and often encounter problems during training. The depth of the model is relatively large, which greatly affects the performance of the system because a large number of layers usually forces the system to memorize. Another limitation of deep networks is the diminishing gradients in the weight matrix. This phenomenon occurs when the parameters in the first layers are updated very slowly, whereas the parameters in the last layers are updated too quickly. Furthermore, in deep networks, the information in the first layers can easily be lost during the forward propagation process and may not contribute anything to generating the output.

The residual mechanism was first introduced in 2015 [24] and was applied to neural networks for image recognition problems. It has been shown that using residual blocks enables the training of deep networks and thus achieves better performance. A key feature of this mechanism is that skip concatenations add the input of a block to the output of that block itself, thereby spreading information throughout the network. In our model, the residual block has the structure shown in Figure 4.

Two regular convolutions are applied to the input feature map, and the output is concatenated with the original input feature map along the channel dimension. A max-pooling operator is then applied to reduce the size.

### 3.2. Dose–Volume Histogram (DVH)-Based Loss Function

To combine deep learning with domain knowledge for radiotherapy dose predictions, we propose a loss function based on the mean absolute error and dose–volume histogram, enabling the model to learn and provide meaningful medical results.

A dose–volume histogram is a histogram of the radiotherapy dose to a tissue volume in a radiotherapy treatment plan [34]. A DVH is often used to evaluate planning and compare doses of different plans. A DVH summarizes the three-dimensional dose distribution into a two-dimensional form. The “volume” in DVH analysis refers to a target of radiation treatment (tumor), a healthy organ near the target, or an arbitrary structure; the DVH represents the dose distribution of the target or organ. In a dose–volume histogram, the column height represents the volume of a target or organ receiving a specific dose, as indicated by the corresponding bin on the histogram. The horizontal axis represents the dose values for each bin, while the vertical axis represents the volume of the target or organ (either as a percentage or an absolute volume). The DVH illustrates the distribution of doses across voxels within a given range, as well as the minimum and maximum doses received by the target or organ.

The DVH-based loss function is constructed as follows: from a dose predicted by the model and its true dose, a set of values derived from the corresponding dose–volume histograms of the predicted and true doses are calculated. These values are called the DVH criteria. The DVH criteria include a series of dose indicators calculated for organs at risk (OARs) and planning target volumes (PTVs). For each organ at risk (OAR), the DVH criteria are calculated as follows: $D_{1}^{i},D_{2}^{i},D_{3}^{i}$, …, $D_{99}^{i}$ are the dose values received by 1% (99th percentile), 2% (98th percentile), 3% (97th percentile), …, 99% (1st percentile) of the number of voxels in the $i^{th}$ organ at risk. Similarly, for the PTVs, the DVH criteria are calculated as follows: $D_{1}^{t},D_{2}^{t},D_{3}^{t}$, …,$D_{99}^{t}$ are the dose values received by 1% (99th percentile), 2% (98th percentile), 3% (97th percentile), …, 99% (1st percentile) of the number of voxels in the target region *t*. As mentioned above, each patient has seven organs at risk and three PTVs, so there is a maximum of 7×99+3×99 = 990 DVH criteria; the number of DVH criteria may be less than 990 because some patients lack information about a certain OAR or PTV. The DVH-based loss function is defined as the mean absolute error between the DVH criteria of the predicted and true doses:$$L_{DVH}\left(D_{p},\hat{D_{p}}\right)=\frac{1}{n_{p}}\sum_{c}\left|\tilde{DVH_{c}}\left(D_{p}\right)-\tilde{DVH_{c}}\left(\hat{D_{p}}\right)\right|,$$
where $L_{DVH}$ is the DVH-based loss function, $D_{p}$ is the true dose of patient *p*, $\hat{D_{p}}$ is the predicted dose of patient *p*, $n_{p}$ is the number of possible DVH criteria for patient *p*, and $\tilde{DVH_{c}}$ is one of the 990 DVH criteria mentioned above.

Our proposed convolutional neural network includes Models A and B, and each model predicts its own dose. Since Model A predicts a coarse dose for the input of Model B, we use a loss function based on the MAE for Model A. The output of Model B is the final predicted dose. We use the DVH-based loss function described above for Model B. The sum of the two loss functions for Models A and B is the total loss for the model. The *Total Loss* for patient *p* is calculated using the following formula:$$Total\;Loss=0.5\times \frac{1}{V_{p}}\sum_{i=1}^{V_{p}}\left|D_{p}\left(i\right)-\hat{D_{p}^{A}}\left(i\right)\right|+L_{DVH}\left(D_{p},\hat{D_{p}^{B}}\right),$$
where $V_{p}$ denotes the total number of voxels that can receive a dose for patient *p*, $D_{p}$ is the true dose for patient *p*, $\hat{D_{p}^{A}}$ is the predicted dose from Model A for patient *p*, and $\hat{D_{p}^{B}}$ is the predicted dose from Model B for patient *p*. Because the output of Model A is designed to support and reduce difficulties during the training process and since this is not the final output, a weight of 0.5 is assigned. In addition, the number of feature maps in Model A is only half that of Model B. The output of Model B is more important, so its weight is larger, and training focuses more on this output.

## 4. Experiment

### 4.1. Setup and Configuration

Our proposed model was built with the Pytorch framework, and the training process was performed on Nvidia T4 Tensor Core GPUs on the Google Colab Pro platform. The experimental process diagram can be observed in Figure 5. In the training process, before being fed into the model, the CT images were standardized with a mean of 919.39 and a standard deviation of 396.02. Both these values were calculated only on the training set to avoid data leakage. The training data were augmented using two methods independently: random flipping along all three axes with a probability of 0.5 for each axis, and image translation with a random distance along all three axes, with a pre-calculated limit to ensure that the translation does not lose any body part. The input images were augmented before being fed into the model. The optimization process was stopped when the model showed no signs of improvement or overfitting occurred. During the testing or inference processes, the CT images were normalized with two calculated mean and standard deviation values and fed to the trained model to produce the predicted dose.

The Adam optimizer [35] with a momentum of $\beta_{1}=0.9$ and $\beta_{2}=0.999$ was used to minimize the loss function between the predicted and true doses. The initial learning rate was 0.001 and was adapted following the one-cycle cosine annealing scheduler [36] and the number of epochs. The formula for cosine annealing is$$\alpha_{t}=\alpha_{\min}+\frac{1}{2}\left(\alpha_{\max}-\alpha_{\min}\right)\left(1+\cos\left(\frac{T_{cur}}{T}\pi \right)\right),$$
where $\alpha_{\min}$ and $\alpha_{\max}$ are the ranges for the learning rate and $T_{cur}$ accounts for how many epochs have been performed. The maximum number of epochs was set to 100, and each epoch took about 1 h to execute. Because of memory limitations, the batch size used was 1. In order for the model to operate subjectively on the dataset under consideration, the model parameters were trained from scratch and did not take advantage of pretrained models. The initial weights followed a normal distribution. Our model consisted of 36,146,194 parameters, all of which were trainable. The hyperparameters are summarized in Table 1.

During the implementation phase, we took the best model weights obtained after the testing phase and developed an interactive interface that allowed for seamless interaction with the model. The model’s weights were saved in a file with the “.pkl” extension, with a size of 137 megabytes. The application was built using Python’s Tkinter library (version 8.6), a powerful tool specifically designed for creating desktop graphical user interface (GUI) applications. The application was designed to accept as input a directory file containing the relevant data for a specific patient. Upon receiving this input, the application processes the data and returns two important outputs: a 2D slice of the patient’s CT image and the corresponding dose map generated by the model. These images are displayed within the application using the Matplotlib library (version 3.5). To ensure the images are easily interpretable, we employed distinct colormap schemes: the CT slice is rendered using a “grayscale” colormap, which is commonly used for medical imaging to highlight structural details, while the dose map is shown with a “turbo” colormap, which provides a vibrant, clear representation of varying dose levels.

### 4.2. Evaluation Metric

The model was evaluated using the *DVH-score*, utilized by the OpenKBP challenge, for the testing dataset [26]. The *DVH-score* is defined as the absolute difference between the corresponding DVH values of the predicted dose and the actual required dose, reflecting the deviation in the dose distribution between the two. For organs at risk (OARs), the two DVH values calculated are the mean dose received by the entire organ (denoted as Dmean) and the maximum dose delivered to the 0.1 cm^3^ of the organ (denoted as D0.1cc). For the planning target volumes (PTVs), the radiotherapy dose received by the volume of each target area is determined at specific volume thresholds. These thresholds are represented by three key volume rates: 1%, 95%, and 99%, denoted as D1, D95, and D99, respectively. Each patient has seven organs at risk and three PTVs, so there are a maximum of 7 × 2 + 3 × 3 = 23 DVH values for each patient (a vector of length 23). The *DVH-score* is a clinically standard metric used to assess the quality of predicted radiotherapy dose distributions. It quantifies the agreement between the predicted and actual dose distributions. The formula for calculating the *DVH-score* is as follows:$${DVH-score}_{p}={∥D\left(s_{p}\right)-D\left({\hat{s}}_{p}\right)∥}_{1},$$
where *p* is the patient, ${\hat{s}}_{p}$ is the predicted radiotherapy dose for patient *p*, $s_{p}$ is the true radiotherapy dose for patient *p*, $D(s_{p})$ is the DVH value of $s_{p}$, $D({\hat{s}}_{p})$ is the DVH value of ${\hat{s}}_{p}$, and ${∥\cdot ∥}_{1}$ is the L1-norm of the vector. With this formula, it can be seen that the smaller the *DVH-score*, the better. The *DVH-score* for multiple patients is the average of the DVH scores for those patients.

## 5. Results

The chart in Figure 6 shows the optimization process of our model, where the value of the loss function on the validation set gradually decreases with the number of epochs. This proves that the proposed DVH-based loss function is effective. Over the first 50 epochs, the loss function on the validation set decreases clearly but is unstable. Over the next 50 epochs, the loss function becomes more stable and smoother, but the decrease rate is slower. Additionally, the loss function on the training set decreases only moderately after 100 epochs, so the optimization process was stopped to limit the overfitting phenomenon.

Figure 7 and Figure 8 illustrate the loss curves for Models A and B separately throughout the entire training process. From these figures, it is clear that both models successfully converged, which is evidence of the effectiveness of the cascade learning mechanism. This convergence indicates that the parameters in each part of the models effectively learned the data’s features and contributed to the overall performance. Furthermore, the convergence rate of both models was relatively fast, suggesting that the models learned efficiently and that the issue of vanishing gradients was well managed, allowing for stable learning over time. However, for Model B, the loss values on the validation set initially showed some instability during the first 60 epochs. This instability was likely due to the model adjusting its parameters and learning rate during the early stages of training. After this initial phase, the loss values became more stable and continued to improve, indicating that the model had successfully adapted and was now learning in a more stable manner.

To illustrate the effectiveness of the residual connection, we trained the model with and without applying the residual connection. The results were calculated on the test set for each region of interest, as summarized in Table 2. For the three target areas, the DVH score was better for the model with the residual connection than for the one without it, with PTV56, PTV63, and PTV70 scoring 1.26, 1.69, and 1.3, respectively. For the seven organs at risk, the model with the residual connection achieved better DVH scores for the brainstem, right parotid, left parotid, and larynx, with values of 1.63, 1.39, 1.41, and 1.66, respectively. The DVH scores were equal for the larynx (2.14) and were only slightly worse for the spinal cord and mandible, with values of 1.18 and 1.54, respectively. The overall DVH score of the model with the residual connection was 1.44, which was better than that of the model without the residual connection. This demonstrates that a residual connection helps preserve information in a deep network, thereby making the inference process achieve better results.

Figure 9 shows the difference between the predicted and ground-truth DVH values of our model on the test set. The medians of all metrics were distributed between −1.036 and 0.536 Gy, and the means were distributed between 0.583 and 0.660 Gy. This shows that the predicted and ground-truth DVH values were relatively close. There were a total of 85 outliers across all DVH values, but only 12.94% of them (about 11 outliers) had a dose difference > 10 Gy or < −10 Gy. The smallest difference was 0 Gy, and the largest difference was just 25 Gy. With our proposed loss function, the model can be trained to minimize the difference between the predicted and ground-truth DVH values.

To provide an objective assessment, the DVH score of our model was compared with that of other models on the testing dataset. These models were divided into two groups. The first group (denoted as 1) included models taken from the OpenKBP competition, such as C3D [37], a U-Net-based model that achieved a good DVH score; 3D dense dilated U-Net [38], a U-Net network whose bottleneck is a DenseNet network; and U-Net-ResNet3D [39], a U-Net with additional ResNet blocks between the up and down convolutions and trained with a feature-based loss. The second group (denoted as 2) included cutting-edge models for similar or nearly similar problems, such as DeepDose [40], the first U-Net model used for dose prediction; HD-U-Net [41], a 3D U-Net with dense hierarchical connections for dose prediction; 2D DCNN [42], a 2D U-Net with dense connections and dilated convolutions applied for dose prediction; Swin-U-Net [43], a 2D U-shaped architecture based entirely on transformers, primarily used for medical image segmentation; and TrDosePred [44], a U-Net built with convolutional patch embedding and multiple local self-attention-based transformers.

The DVH-scores for the different models are presented in Table 3, providing a comprehensive comparison of their performance. Our model achieved a DVH score of 1.444 Gy, which corresponded to 2.06% of the prescribed dose for the planning target volume (PTV70). This result demonstrates that our model not only delivers accurate dose predictions but also performs better than the other models listed in the table, highlighting its superior effectiveness in predicting radiation doses in comparison to alternative approaches.

Table 4 compares the specific metrics included in the DVH score for four models on the test set: DeepDose, HD-U-Net, TrDosePred, and ours. DeepDose, HD-U-Net, and TrDosePred are ensemble models. Our model predicted $D_{99},D_{95},$ and $D_{1}$ within 1.472 ± 2.184 Gy, 1.181 ± 1.816 Gy, and 1.407 ± 1.238 Gy, and although it predicted $D_{mean}$ and $D_{0.1cc}$ within 1.306 ± 1.405 Gy and 1.704 ± 2 Gy, all five indices outperformed those of the other three models. In addition, the standard deviation values of our model were relatively small, demonstrating that the prediction results were highly stable.

In order to conduct a comprehensive demonstration and thoroughly evaluate the performance of the proposed model across different scenarios, we randomly selected three samples from the test dataset, specifically patients 274, 279, and 313. This selection was made to assess how the model performed on a range of individual cases within the dataset. The outcomes of these evaluations, which provide insight into the model’s performance in each of these cases, are presented in Figure 10 and Figure 11, and Table 5.

Table 5 presents the DVH score as well as the values of D99, D95, D1, Dmean, and D0.1cc for the three tested patients. Patient 279 had the best DVH score with a value of 0.741, followed by patient 313 with a DVH score of 0.954, and finally, patient 274 with a DVH score of 1.229.

Figure 10 shows a comparison of the ground-truth and predicted DVH curves for the three patients on the testing dataset. The solid lines represent the ground-truth dose distributions, and the dashed lines represent the predicted dose distributions of our model. The charts for patients 274, 279, and 313 are arranged in order from top to bottom, corresponding to each patient’s respective chart. Since each patient entry may not include information on certain RoIs, the number of DVH curves varies. As can be seen in the charts, the dashed lines and solid lines almost overlap for all three patients, which means that the predicted dose was close to the true dose. For patient 274, the doses for the larynx, esophagus, and brainstem regions are relatively similar, as their corresponding DVH curves are very close to each other. The DVH curves for patient 279 are relatively overlapped, with the exception of the left parotid and right parotid, which show some deviation in certain sections. For patient 313, the DVH curves for the OAR regions are quite close to each other, while the curves for the PTV regions only deviate slightly.

Figure 11 provides a detailed visualization of the 3D dose distributions for the three patients. The first column displays the CT images of the patients, offering a clear view of the anatomical structures, while the second column illustrates the dose distributions predicted by our model, showing how the radiation is distributed across the body. The third column presents the ground-truth dose distributions, representing the actual radiation doses delivered. In the dose maps, the intensity of the radiation dose is visually encoded with colors, where red indicates higher radiation doses and cooler colors signify lower doses. Upon visual inspection, it is evident that the predicted dose distributions closely align with the ground truths, particularly in regions with high dose concentrations. These high-intensity areas are primarily located within the planning target volumes (PTVs), which are the areas identified for precise radiation delivery, further highlighting the model’s accuracy in predicting the dose distribution in critical regions.

Table 6 shows the effect of the components in our proposed method. To evaluate the performance of the dual-model architecture, we conducted further experiments with the single-model case. The model chosen in the single-model architecture was Model B because this is the network that predicts the dose directly. It can be seen that when using the same loss function, the mean absolute error, the dual-model architecture achieved better results than the single model for most indices, except for $D_{mean}$, where it was slightly worse. The dual model was trained more than the single model in a single epoch because it consisted of two consecutive networks with two separate loss functions. In addition, the second model was provided with the raw dose from the first model, so it only needed to fine-tune that raw dose to obtain a better final dose. This was not true for the single model, where its input only included information about the CT image and the OAR regions and had to predict the final dose. Next, we trained two dual-model experiments with two loss functions, the mean absolute error and the DVH-based loss, to evaluate the effectiveness of our proposed loss function. It can be seen that the results were significantly better across all metrics when the model was trained with the DVH-based loss function. Instead of optimizing based on each voxel like the MAE function, our loss function optimizes based on the % concentration in each OAR area, thereby making the predicted dose more meaningful from a medical perspective. This is shown through the formula of the DVH-based loss function.

Finally, to enhance the accessibility and practical application of our proposed model, we developed a software tool designed to facilitate the prediction of radiation doses for patients, as well as to provide an intuitive interface for visualizing the resulting dose maps. Figure 12 illustrates the software’s user interface.

The patient selection process is straightforward, allowing users to select a patient using the “Choose a patient” button. It is important to note that only one patient can be selected at a time to ensure clarity and focus on the individual patient’s dose distribution. We tested the software on a standard computer equipped with an Intel i5-8250U processor and without a dedicated GPU of Acer. Despite these modest specifications, the software demonstrated impressive performance, taking only about 5 s from the moment the “Choose a patient” button was clicked until the predicted dose map and results were displayed. Once the results are loaded, the user is presented with a 2D slice of the patient’s CT image, along with the corresponding radiation dose map, making it easy to observe how the radiation is distributed across the body. Additionally, the software offers flexibility in navigation, as users can select specific dimensions and slice numbers, allowing them to explore and examine the precise slice of interest, along with its corresponding dose distribution. This capability makes it convenient for users to view and analyze the radiation dose in various parts of the body, ensuring that they can focus on the most relevant regions based on their clinical needs.

## 6. Conclusions

In this study, we present a custom-designed convolutional neural network (CNN) tailored for the task of predicting radiotherapy doses. The architecture of the model incorporates two successive U-Net variants, which are designed to effectively capture spatial relationships within medical images. To optimize the model’s performance, we introduce a novel loss function based on the dose–volume histogram (DVH), which is utilized during the training process. This loss function allows the model to more accurately predict the radiotherapy dose distribution by comparing the predicted dose with the true dose values. For the experimental evaluation, we employed data from head and neck cancer patients who underwent radiation therapy, sourced from the Open Knowledge-Based Planning Challenge (OpenKBP). This dataset provides a robust and varied set of images and dose information, making it ideal for testing the model’s predictive accuracy. Prior to the training phase, several data augmentation and preprocessing techniques were applied to the dataset to improve model generalization and robustness. These techniques included standardization, image translation, and flipping, which helped introduce diversity in the data and prevent overfitting during the training process. Furthermore, our proposed method can be generally applied to data represented as a three-dimensional matrix with dimensions other than 128×128×128, and the number of organs at risk and PTV can vary. This can be done by changing a small part of the configuration in the network architecture, as well as the formula of the proposed loss function.

The results of our experiment reveal that the model’s predicted radiotherapy dose is remarkably close to the actual dose, as illustrated in the charts and visual representations presented in the preceding sections. This indicates that our custom CNN is effective in predicting dose distributions with high accuracy. Many studies have also been conducted and published using the OpenKBP dataset, yielding remarkable results. Zimmermann et al. [39] employed a 3D U-Net model enhanced with extra ResNet blocks in both the encoder and decoder. They used an L1 (MAE) loss in conjunction with a feature loss, where a pre-trained video classifier served as the feature extractor. Liu et al. [37] proposed a model named C3D and used the mean absolute error function during training. Gronberg et al. [38] experimented with several network architectures, with their best-performing model utilizing a 3D U-Net. This model featured a dilated DenseNet block inserted between the encoder and decoder, a weighted MSE loss function, and a patch-based strategy. Our model’s performance, measured using the DVH score, exceeds that of other models included in the OpenKBP challenge, as well as several advanced, state-of-the-art models tackling similar dose prediction problems in the medical imaging domain. This performance demonstrates the effectiveness of the proposed architecture and DVH-based loss function in addressing the complex task of radiotherapy dose prediction. In addition to the model itself, we developed a user-friendly software tool to facilitate the practical application of the model’s predictions. This software is designed to display the predicted dose maps for each 2D slice of the CT images, providing a visual representation of the dose distribution. The tool allows users, including clinicians and researchers, to easily interact with the model’s output, offering a convenient platform for viewing and analyzing the predicted dose maps. The software’s intuitive interface makes it accessible for use in clinical settings, helping users assess and interpret the predicted dose distributions efficiently. Through this comprehensive approach, we aim to enhance the usability of our model and contribute to improving radiotherapy planning and treatment in clinical practice.

A limitation of this study is the size of the dataset used. Deep learning methods typically perform better with larger datasets. While our proposed model did not exhibit overfitting, as the validation loss decreased alongside the training loss, having more data could reduce the gap between the training and validation loss curves. As can be seen in the ablation study, the use of a dual model yielded better results than the use of a single model. This improvement is due to the second model using additional output information from the first model to predict the final dose. However, the study did not show how the output information from the first model contributed to and specifically affected the final prediction result.

Cancer has long been one of the most challenging problems facing humanity, and despite significant advancements, it continues to present considerable difficulties. Researchers around the world are tirelessly working to improve treatment methods, aiming to develop more effective and targeted therapies. One critical aspect of cancer treatment is determining the appropriate radiation dose for patients, a process that is both time-consuming and heavily reliant on the expertise of medical professionals. This reliance on expertise often results in higher treatment costs for patients, as radiation planning can be complex and labor-intensive. In response to this challenge, we have developed a deep learning model that demonstrates promising performance in predicting radiation doses. The model, while effective, is still in its early stages and requires further refinement to meet the demanding standards of real-world medical applications. In the future, we plan to develop a program that is capable of displaying 3D predictions of radiation doses instead of 2D slices, as shown above. By providing 3D visualizations of the dose distributions, we aim to offer a more comprehensive, intuitive, and interactive experience for physicians and other healthcare providers. This advancement will not only make it easier for users to understand the spatial relationships in the radiation dose but also improve the overall efficiency of treatment planning. Furthermore, we will attempt to test on a new dataset or combine datasets to train it in the most general way across the whole body. Additionally, we will conduct further research on cascade learning mechanisms in deep learning and investigate how each component contributes to the final prediction result, as well as whether combining state-of-the-art models could create a new state-of-the-art model. With these enhancements, we hope to significantly contribute to the ongoing efforts to improve cancer treatment, making the process more efficient and accessible to both medical professionals and patients.

## Figures and Tables

**Figure 1 diagnostics-15-00177-f001:**
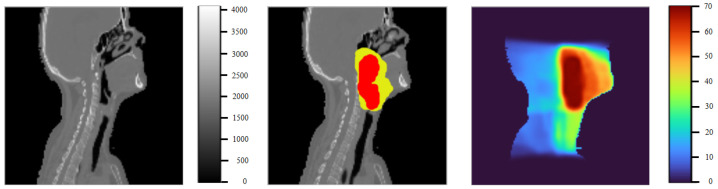
Illustration of a 2D slice image of a patient. The first image is a CT image, the second image contains information about the PTV areas, and the last image is the corresponding radiation therapy dose.

**Figure 2 diagnostics-15-00177-f002:**
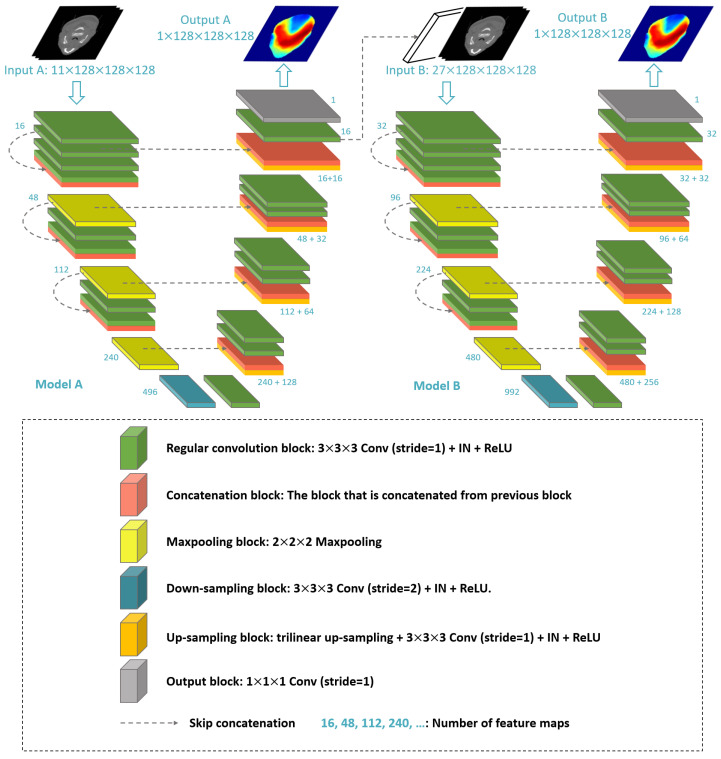
The architecture of our proposed model.

**Figure 3 diagnostics-15-00177-f003:**
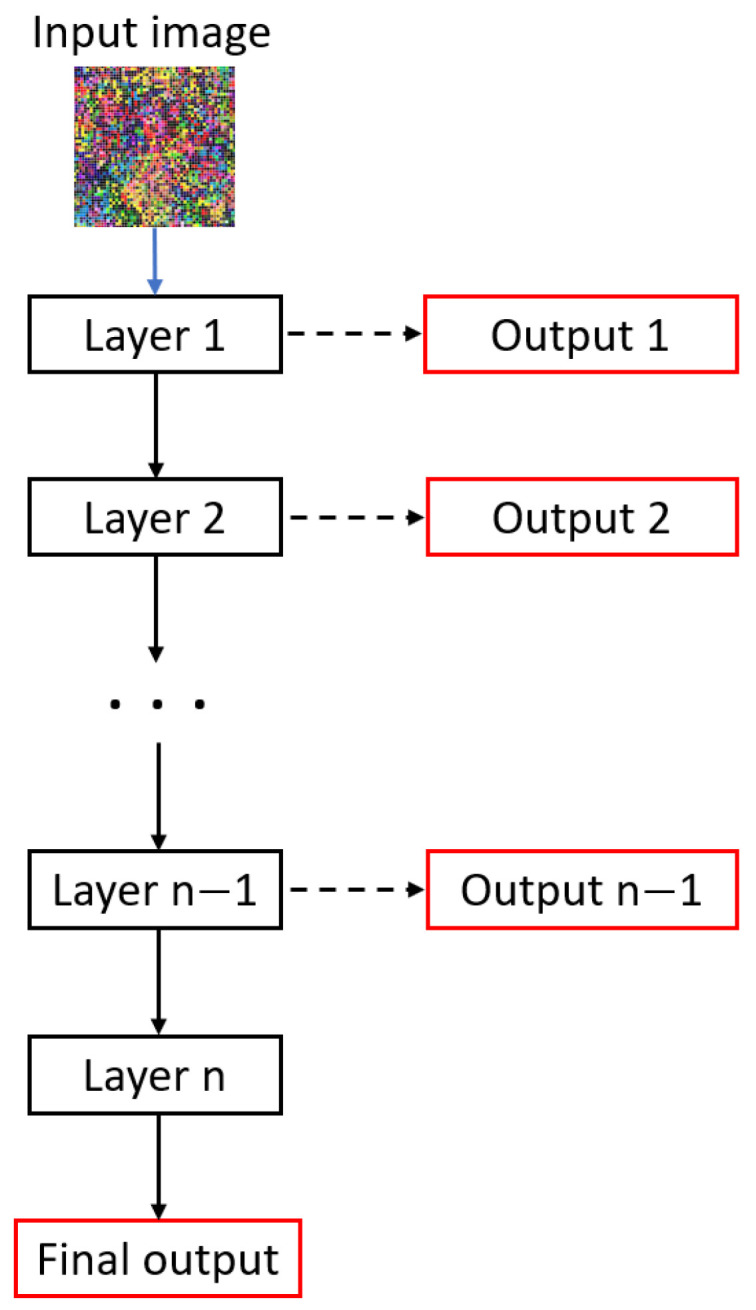
Overview of cascade learning in deep learning.

**Figure 4 diagnostics-15-00177-f004:**
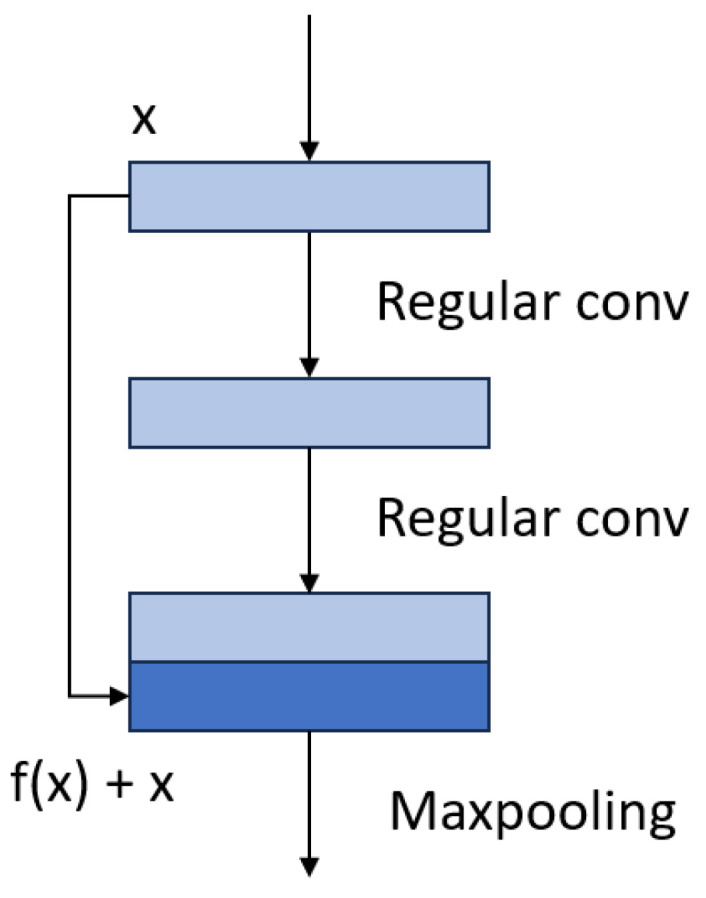
Residual block.

**Figure 5 diagnostics-15-00177-f005:**
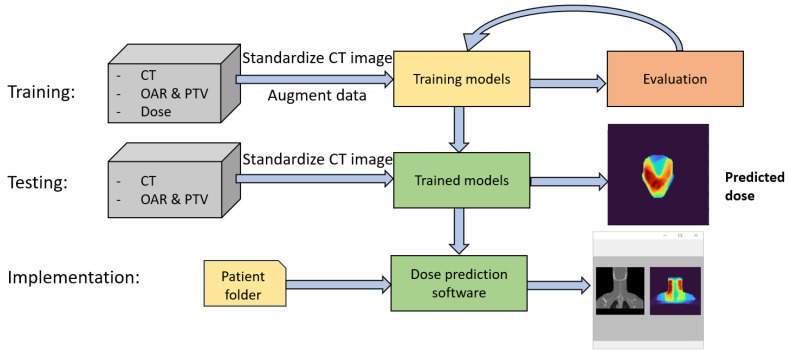
Flowchart representing the training, testing, and implementation phases.

**Figure 6 diagnostics-15-00177-f006:**
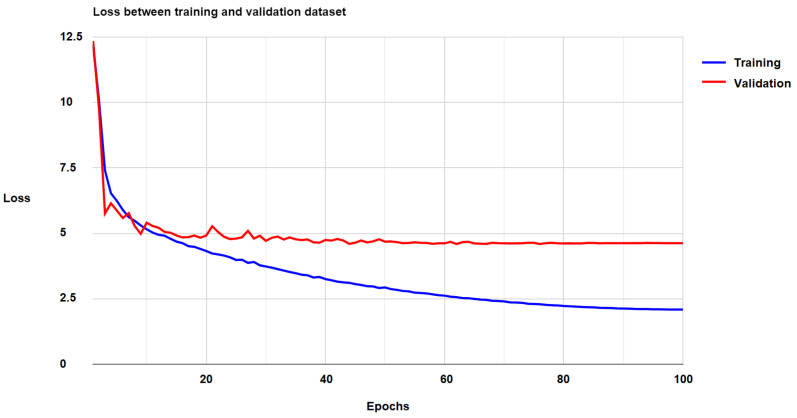
The total loss of our model on the training and validation datasets.

**Figure 7 diagnostics-15-00177-f007:**
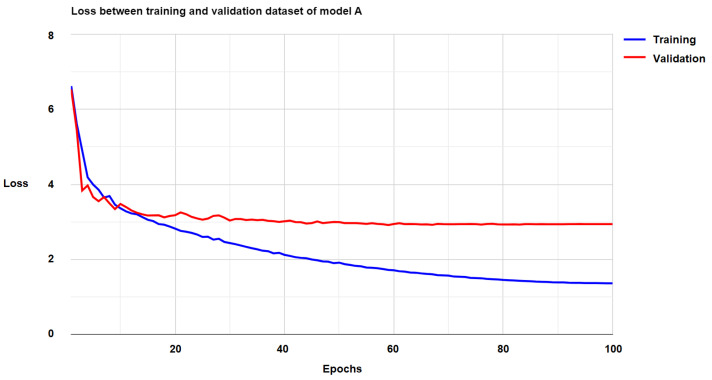
The loss of Model A on the training and validation datasets.

**Figure 8 diagnostics-15-00177-f008:**
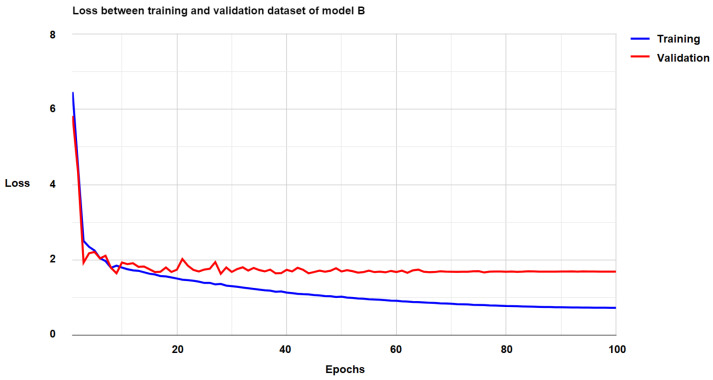
The loss of Model B on the training and validation datasets.

**Figure 9 diagnostics-15-00177-f009:**
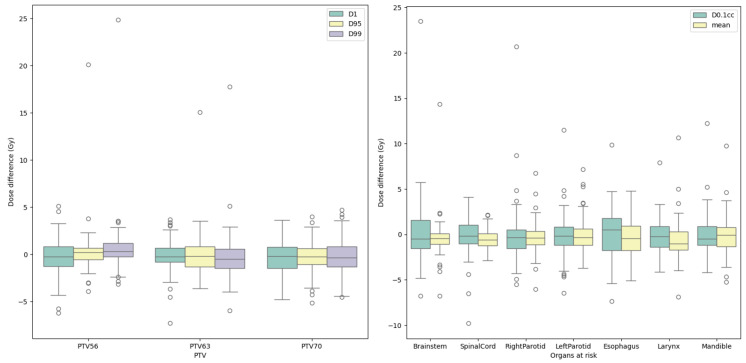
The difference between the predicted and ground-truth DVH values of our model on the test set.

**Figure 10 diagnostics-15-00177-f010:**
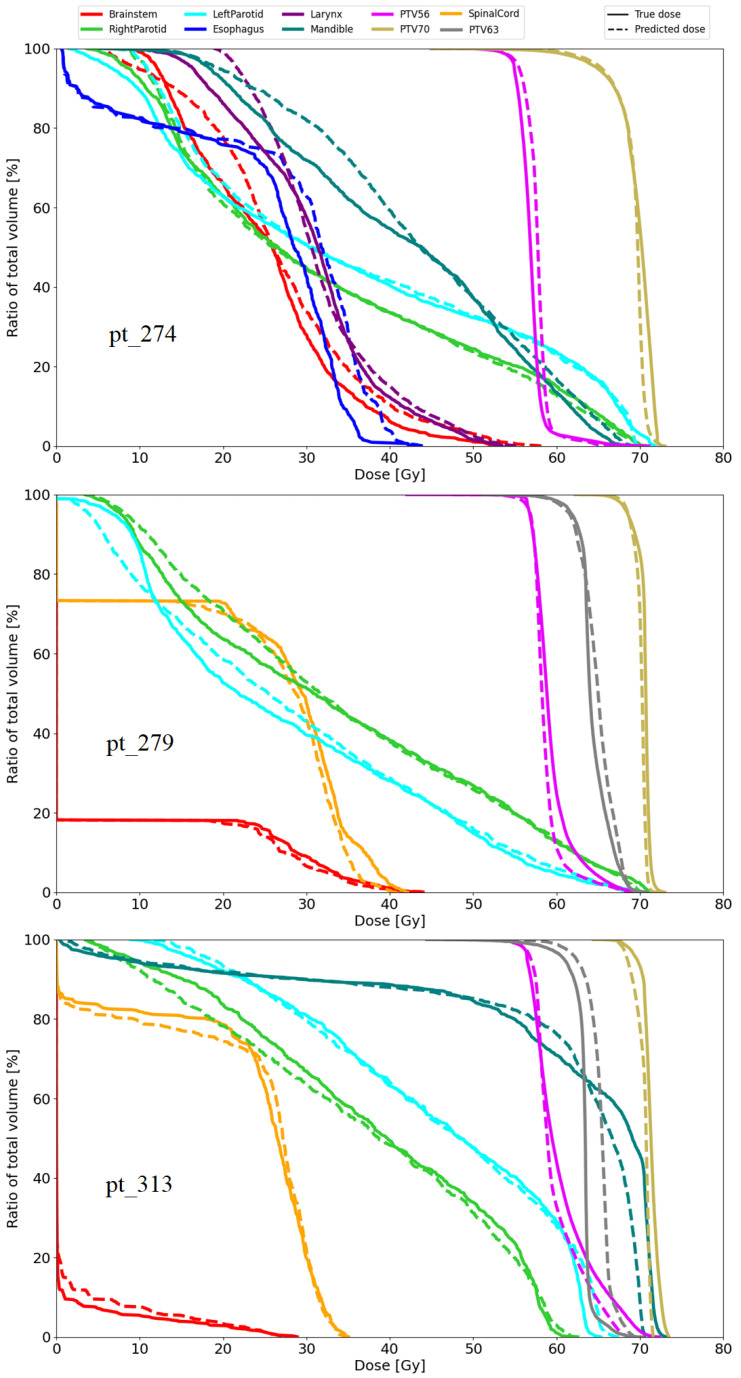
Comparison of the predicted (dashed lines) and ground-truth (solid lines) dose–volume histograms for three patients: 274, 279, and 313.

**Figure 11 diagnostics-15-00177-f011:**
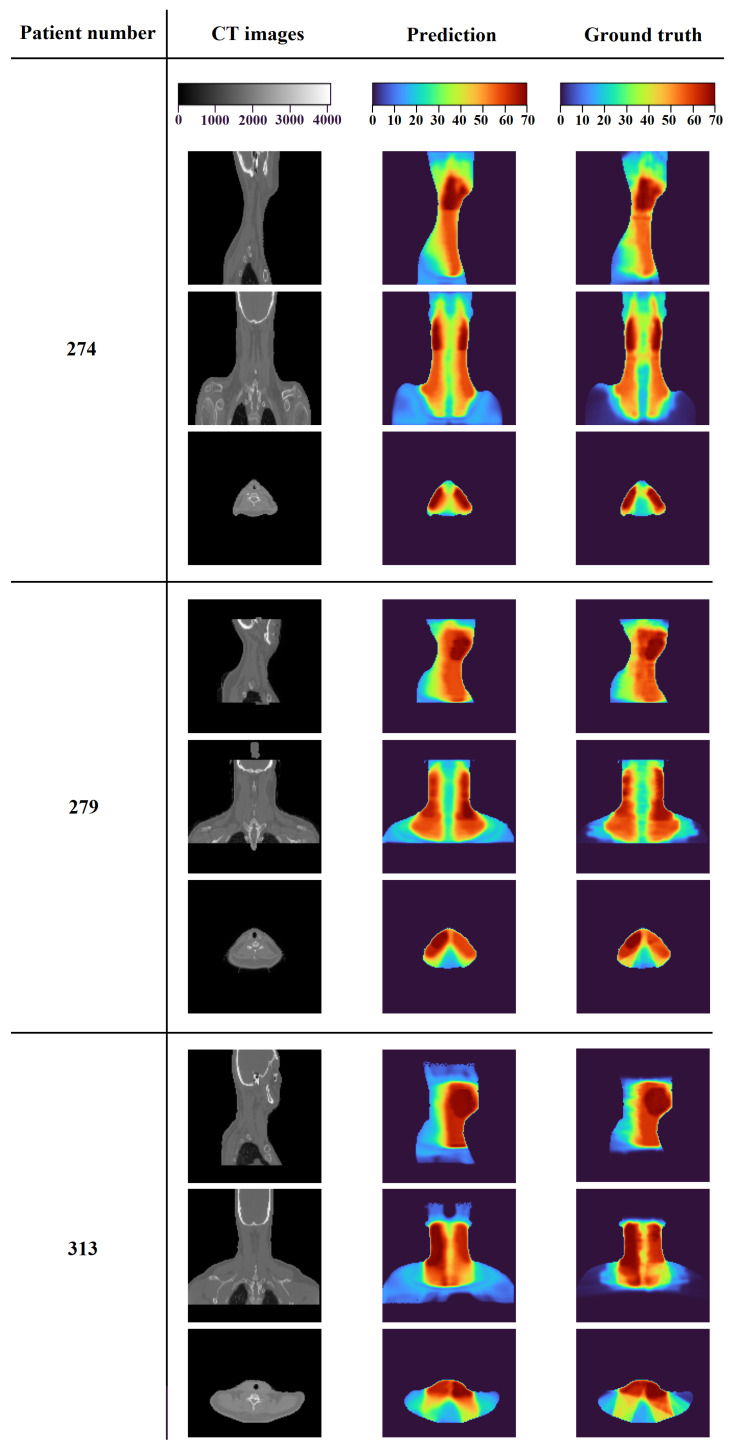
Three-dimensional dose distributions for three patients: 274, 279, and 313.

**Figure 12 diagnostics-15-00177-f012:**
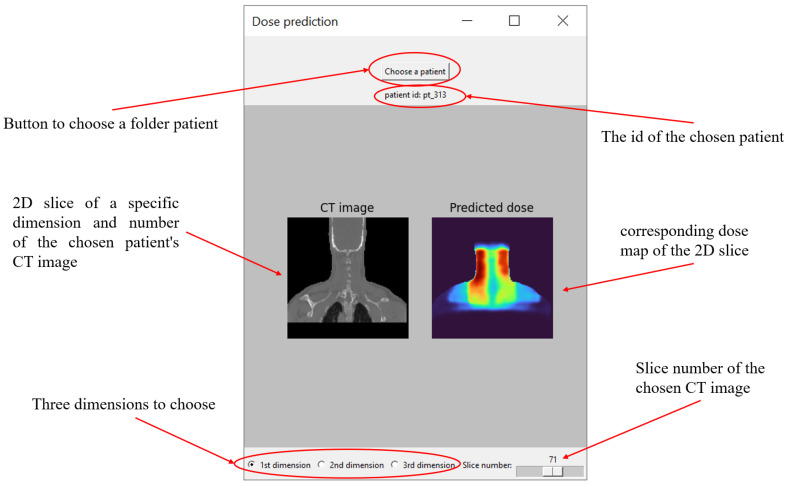
The interface of the software for predicting the radiation dose.

**Table 1 diagnostics-15-00177-t001:** Configuration of training hyperparameters for the proposed model.

Optimizer	Adam
Initial learning rate	0.001
Momentum	0.9, 0.999
Learning rate schedule	Cosine annealing
Epochs	100
Batch size	1
Input image size	128 × 128 × 128
Params	36, 146, 194

**Table 2 diagnostics-15-00177-t002:** The *DVH-score* at each RoI (region of interest) for the model with and without applying a residual connection on the test set.

RoIs	With Residual	Without Residual
Brainstem	**1.63**	1.76
Spinal Cord	1.18	**1.17**
Right Parotid	**1.39**	1.62
Left Parotid	**1.41**	1.45
Esophagus	**2.14**	**2.14**
Larynx	**1.66**	1.70
Mandible	1.54	**1.50**
PTV56	**1.26**	1.27
PTV63	**1.69**	1.71
PTV70	**1.30**	1.33
Overall	**1.44**	1.50

**Bold** indicates better value.

**Table 3 diagnostics-15-00177-t003:** Comparison of DVH scores for various models on the test set.

Model	DVH Score (Gy)
C3D ^1^	1.478
3D DCNN ^1^	1.704
U-Net-ResNet3D ^1^	1.582
DeepDose ^2^	1.741
HD-U-Net ^2^	1.802
2D DCNN ^2^	1.620
Swin-U-Net ^2^	1.757
TrDosePred ^2^	1.592
Ours	1.444

^1^ the models taken from the OpenKBP competition. ^2^ the cutting-edge models for similar or nearly similar problems.

**Table 4 diagnostics-15-00177-t004:** Comparison of DVH metrics for various models on the test set (mean ± standard deviation).

Model	$D_{99}$ (Gy)	$D_{95}$ (Gy)	$D_{1}$ (Gy)	$D_{\mathrm{mean}}$ (Gy)	$D_{0.1\mathrm{cc}}$ (Gy)
DeepDose	2.001 ± 2.465	1.494 ± 2.003	1.777 ± 1.419	1.410 ± 1.527	1.894 ± 2.162
HD-U-Net	2.023 ± 2.436	1.579 ± 2.028	1.774 ± 1.342	1.323 ± 1.465	1.894 ± 2.157
TrDosePred	1.838 ± 2.383	1.407 ± 1.964	1.474 ± 1.269	1.312 ± 1.442	1.898 ± 2.185
Ours	**1.472** ± 2.184	**1.181** ± 1.816	**1.407** ± 1.238	**1.306** ± 1.405	**1.704** ± 2

**Bold** indicates better value.

**Table 5 diagnostics-15-00177-t005:** The DVH metrics for three patients: 274, 279, and 313.

Patient Number	$D_{99}$ (Gy)	$D_{95}$ (Gy)	$D_{1}$ (Gy)	$D_{\mathrm{mean}}$ (Gy)	$D_{0.1\mathrm{cc}}$ (Gy)	DVH-Score
274	0.913	1.074	0.649	1.261	1.548	1.229
279	0.935	0.661	0.167	0.561	1.265	0.741
313	1.084	0.611	1.720	0.453	1.122	0.954

**Table 6 diagnostics-15-00177-t006:** Comparison with ablation models.

Model	$D_{99}$ (Gy)	$D_{95}$ (Gy)	$D_{1}$ (Gy)	$D_{\mathrm{mean}}$ (Gy)	$D_{0.1\mathrm{cc}}$ (Gy)	DVH Score
Single model + MAE loss	1.918	1.478	1.669	1.382	2.105	1.722
Dual model + MAE loss	1.740	1.333	1.634	1.455	2.046	1.679
Dual model + DVH-based loss	**1.472**	**1.181**	**1.407**	**1.306**	**1.704**	**1.444**

**Bold** indicates better value.

## Data Availability

The OpenKBP dataset is available at https://github.com/ababier/open-kbp (accessed on 15 October 2024).

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
