# Peer review of "An Efficient 3D Convolutional Neural Network for Dose Prediction in Cancer Radiotherapy from CT Images"

_diagnostics, 2025, doi:10.3390/diagnostics15020177_

Round 1

Reviewer 1 Report

Comments and Suggestions for Authors

The manuscript presents a novel approach to dose prediction in cancer radiotherapy using 3D convolutional neural networks (CNNs). The manuscript is well-structured, with a clear introduction, methodology, results, and conclusions. The authors have provided a comprehensive analysis of their approach, including a comparison with existing methods. However, there are several aspects where the manuscript could be improved.

1. The paper mentions the use of two 3D U-Net models (Model A and Model B), but does not provide a detailed explanation of why this dual-model architecture was chosen instead of a single model or other architectures. Please provide a detailed comparison of the dual-model architecture with a single model in terms of performance and explain in which aspects the dual-model architecture performs better.

2. Ablation experiments are necessary. Please conduct ablation experiments on the improvements mentioned in the paper, such as the loss function and the dual-model architecture.

3. Reproducibility of Experimental Results: The paper does not mention the reproducibility of the experimental results, particularly whether these results can be replicated under different datasets or experimental conditions. Please discuss the reproducibility of the experimental results and explain whether these results can be replicated under different datasets or experimental conditions. In addition, it is better to provide in detail the radiation therapy techniques used for treating the head and neck cancer patients.

4. The weight for the Total Loss is assigned to 0.5. Although the author explain why it is set smaller than the weight of model B, its range of choices is still too wide. What impact changing this value to other values would have on the model's performance and is there a suitable range for its setting.

Author Response

Dear Professors,

We would like to thank Professors, who have spent the time to read and comment on our manuscript.

All comments from reviewers are revised and highlighted to the manuscript

Our following answers may not yet meet your expectations, but we hope you will consider, support, and accept our works.

Kind Regards,

Nang-Toan Do

Associate Professor, Ph.D.

---------------------------------------------------------------------------------------------------------------

Reviewer 1

Comment 1: The paper mentions the use of two 3D U-Net models (Model A and Model B), but does not provide a detailed explanation of why this dual-model architecture was chosen instead of a single model or other architectures. Please provide a detailed comparison of the dual-model architecture with a single model in terms of performance and explain in which aspects the dual-model architecture performs better.

Response: We noticed that the content of comment 1 was already covered in comment 2. Therefore, we added more comparisons and explanations between dual-model and single model architectures in the ablation study section.

Comment 2: Ablation experiments are necessary. Please conduct ablation experiments on the improvements mentioned in the paper, such as the loss function and the dual-model architecture.

Response: We have conducted further ablation experiments to demonstrate the effectiveness of our proposed components. The ablation experiments are presented in the near-end of Section 5 Results, including the proposed loss function and the dual-model architecture.

Revised lines: 435-452

Comment 3: Reproducibility of Experimental Results: The paper does not mention the reproducibility of the experimental results, particularly whether these results can be replicated under different datasets or experimental conditions. Please discuss the reproducibility of the experimental results and explain whether these results can be replicated under different datasets or experimental conditions. In addition, it is better to provide in detail the radiation therapy techniques used for treating the head and neck cancer patients.

Response: The dataset we used is about head and neck cancer patients. Only in this area, there are many types of cancer such as: oral cancer, nasopharyngeal cancer, laryngeal cancer, thyroid cancer, salivary gland cancer, skin cancer, tonsil cancer, ... However, using diverse data will still give a more comprehensive result. Therefore, we have mentioned this issue in the limitation section as well as raised it in the future research section.

We have also added some discussion on the reproducibility of the proposed method under different experimental conditions in the first paragraph of Section 6 Conclusions.

Revised lines: 487-491

We have provided in detail some radiation therapy techniques used for treating the head and neck cancer patients in the second paragraph of Section 2 data preparation.

Revised lines: 109-119

Comment 4: The weight for the Total Loss is assigned to 0.5. Although the author explain why it is set smaller than the weight of model B, its range of choices is still too wide. What impact changing this value to other values would have on the model's performance and is there a suitable range for its setting.

Response: As can be seen in the description of the proposed network architecture in section 3.1, the number of feature maps of model A is half that of model B. Therefore, in the formula of total loss, we also assign the loss of model A with 0.5, which is equal to half that of model B.

Mathematically, when optimizing a function that is a sum of smaller functions, convergence occurs simply by having all the components that make up the function converge. Giving different weights to the components is just like giving priority to the components in the optimization process. The speed of convergence may be different but it does not affect the final convergence. Therefore, in our total loss, the loss of model B is given twice the weight of model A just to give the optimization process a slight preference for model B. Furthermore, looking at the loss graphs of models A and B, it can be seen that both models converge after a certain optimization time.

We have also added some explanation on this issue in Section 3.2.

Revised lines: 287-288

Reviewer 2 Report

Comments and Suggestions for Authors

This paper focuses on using a deep learning model to predict radiation therapy doses for cancer based on CT images. Here are my comments on the given text:

The proposed method is evaluated on a single dataset, which may limit the generalizability of the findings. It would be better to validate the proposed method with another dataset.

Please include a table and list all the hyperparameters of the proposed model in it.

Please add a confusion matrix of the proposed model.

Please add a section to discuss the “limitations of the study.”

In the Conclusion section, elaborate more on future research directions. For instance, you could mention the potential application of the model in different biomedical fields. This section requires further discussion.

Rewrite the Conclusion section by summarizing your method, findings, and comparing them with the papers published. Include a table to facilitate better comparison.

Author Response

Dear Professors,

We would like to thank Professors, who have spent the time to read and comment on our manuscript.

All comments from reviewers are revised and highlighted to the manuscript

Our following answers may not yet meet your expectations, but we hope you will consider, support, and accept our works.

Kind Regards,

Nang-Toan Do

Associate Professor, Ph.D.

---------------------------------------------

Reviewer 2

Comment 1: The proposed method is evaluated on a single dataset, which may limit the generalizability of the findings. It would be better to validate the proposed method with another dataset.

The dataset we used is about head and neck cancer patients. Only in this area, there are many types of cancer such as: oral cancer, nasopharyngeal cancer, laryngeal cancer, thyroid cancer, salivary gland cancer, skin cancer, tonsil cancer, ... However, using diverse data will still give a more comprehensive result. Therefore, we have mentioned this issue in the limitation section as well as raised it in the future research section.

Comment 2: Please include a table and list all the hyperparameters of the proposed model in it.

We have added a summary table of the hyperparameters of the proposed model. This addition is located in Section 4.1 Setup and configuration.

Revised lines: 311

Comment 3: Please add a confusion matrix of the proposed model.

We currently do not understand the confusion matrix concept that the reviewer is referring to. However, we have added an ablation study to our method to demonstrate that the proposed method is effective. The ablation study is included in Table 6 of Section 5 Results. If this is not what the reviewer is referring to, please provide feedback.

Revised lines: 435-452

Comment 4: Please add a section to discuss the “limitations of the study.”

We have added limitations of the study in the middle of Section 6 Conclusions. The added paragraph is highlighted.

Revised lines: 518-526

Comment 5: In the Conclusion section, elaborate more on future research directions. For instance, you could mention the potential application of the model in different biomedical fields. This section requires further discussion.

We have described in more detail the research we plan to do in the future, including further developments of this paper as well as addressing remaining limitations. The contents of these future research are presented in the final paragraph of Section 6 Conclusions.

Revised lines: 543-547

Comment 6: Rewrite the Conclusion section by summarizing your method, findings, and comparing them with the papers published. Include a table to facilitate better comparison.

We have rewritten the contents of the summary of the proposed method, the results obtained and the comparison with previous studies to make them clearer. In addition, we have separated them into separate paragraphs for easier understanding.

In Section 5 Results, we already have two tables (tables 3 and 4) comparing our method with previous studies. Therefore, we think that adding such tables in this section is unnecessary.

Revised lines: 473-517

Reviewer 3 Report

Comments and Suggestions for Authors

The paper examines the dose prediction in cancer radiotherapy from CT images.

Due to extensive overlap with the same authors article published in Techscience, I recommend rejecting the article. 

Major

1. The paper has 6% text overlap with an already published article in https://www.techscience.com/iasc/v39n2/56497/html. The overlap is in the main text and cannot be tolerated at this level. The overlap is a direct citation of about 480 words and phrases from the mentioned article. iThenticate report resulted in 18% match.

2. How much time is used to apply the model in test set per patient? Is this 5 s mentioned in line 431?  Or is these 5 s a comparison of precalculated plans with the ground truth one? This is a relevant question in case of future clinical use.  How much time was used for training on T4 Nvidia Tensor Core system in Google? In line 294 1 hour per epoch is mentioned, but how does it translate per patient or image?

3. How are the train, validation and test patient sets chosen (line 107)? Are they fixed in TCIA database, or does the choice is random and train and validation set is played until saturation of the results, that are then tested in test set?

4. Can you share the program used to generate Figure 12 as a supplementary content?

Minor

1. In line 136nn the units of CT images are mentioned for conversion to miu scale. Do the authors use Hounsfield units from the dataset? 

2. CT images are shown without windows of Hounsfield unit, which makes them very insensitive to small density variations. The typical equivalent in HU is 0…100 HU, that needs to be translated to zero-mean scale used in the paper. 

3. In section 4.2 between lines 313 and 314 the term 0.1cc is used. Does it mean 0.1 cm^3? If so, please use SI units and not cc abbreviation. 

4. What are typical pixel sizes for 128x128x128 initial CT images?

Author Response

Dear Professors,

We would like to thank Professors, who have spent the time to read and comment on our manuscript.

All comments from reviewers are revised and highlighted to the manuscript

Our following answers may not yet meet your expectations, but we hope you will consider, support, and accept our works.

Kind Regards,

Nang-Toan Do

Associate Professor, Ph.D.

---------------------------------------------------------------------------------------------------------------

Reviewer 3

Major

Comment 1: The paper has 6% text overlap with an already published article in
https://www.techscience.com/iasc/v39n2/56497/html. The overlap is in the main text and
cannot be tolerated at this level. The overlap is a direct citation of about 480 words and
phrases from the mentioned article. iThenticate report resulted in 18% match.

Response: The article mentioned by the reviewer in Techscience is one of our group's previous studies and this article is one of the follow-up studies of that article.

We have reviewed the 6% text overlap between the two papers. The duplicated text only covers a small portion of the Introduction, data description, and evaluation metric sections. These are common to both papers and we have revised all of these sections to avoid duplication. We guarantee that there will be no duplication in methods, experiments, results, conclusions, and ideas between the two papers. Reviewers and editors can review our revised manuscript and provide us with any necessary comments.

Revised lines: 1, 34-35, 79-82, 102-108, 120

Comment 2: How much time is used to apply the model in test set per patient? Is this 5 s mentioned in line 431? Or is these 5 s a comparison of precalculated plans with the ground truth one? This is a relevant question in case of future clinical use. How much time was used for training on T4 Nvidia Tensor Core system in Google? In line 294 1 hour per epoch is mentioned, but how does it translate per patient or image?

Response:  When using an i5-8250U CPU (no GPU) computer, the time it takes for the model to be applied to the data on one patient is 5 seconds. When using a device with a GPU, the time it takes to compare the predicted concentration and the ground truth for one patient is about 1.82 seconds. As shown in the data below, it takes about 3 minutes and 2 seconds for 100 patients in the test set.

Each epoch takes about 1 hour to execute. The time it takes us to train 100 epochs is about 100 hours. As shown in the figure below, our training set consists of 800 samples and each sample takes about 4.48 seconds to train (the batch size is is set to 1).

Comment 3: How are the train, validation and test patient sets chosen (line 107)? Are they fixed in TCIA database, or does the choice is random and train and validation set is played until saturation of the results, that are then tested in test set?

Response: They are fixed datasets. This gives us a common training and testing dataset, which we can compare fairly with previously published studies.

Comment 4: Can you share the program used to generate Figure 12 as a supplementary content?

Response: The program source code is available in the following drive link:

https://drive.google.com/drive/folders/1mZ0F7ynAwt7Fo3Y5kGsnCCUaCG2gqiIZ?usp=sharing

The main objective of this paper is to present an efficient method for predicting radiation dose, including the proposed network architecture and loss function. The program is built by us to create an interface for easy manipulation of the proposed method. Therefore, we do not want to attach the program to the supplementary content. However, we are willing to provide the program to any reader in need.

Minor

Comment 1: In line 136nn the units of CT images are mentioned for conversion to miu scale. Do the authors use Hounsfield units from the dataset?

Response: According to the authors who provided the data, the original data used Hounsfield units, then they converted the data to 0-4095 scale. All images in our manuscript are drawn at 0-4095 scale, only when the data is fed into the model during training or testing, the data is normalized with the formula mentioned in the manuscript.

Comment 2: CT images are shown without windows of Hounsfield unit, which makes them very insensitive to small density variations. The typical equivalent in HU is 0…100 HU, that needs to be translated to zero-mean scale used in the paper.

Response: We have added color bars in Figure 1.

Comment 3: In section 4.2 between lines 313 and 314 the term 0.1cc is used. Does it mean 0.1 cm^3? If so, please use SI units and not cc abbreviation.

Response: Yes, 0.1cc means 0.1 cm^3. We have revised this abbreviation in the manuscript.

Comment 4: What are typical pixel sizes for 128×128×128 initial CT images?

Response: As described by the dataset provider, each voxel will be approximately 5mm × 5mm × 3mm in size. Different patients will have different sizes, but the difference is not too large compared to the above figures.

Round 2

Reviewer 1 Report

Comments and Suggestions for Authors

The comments have been  addressed. There are no other comments any more.

Comments on the Quality of English Language

The  English expression is clear.

Author Response

We would like to thank the reviewer’s comments on our research. Thanks to those comments, we were able to revise, supplement and produce a better manuscript.

Reviewer 3 Report

Comments and Suggestions for Authors

The manuscript can be accepted with minor changes, which are the following:

1. Add color bar with units to Figure 11 as it was done for Figure 1.

2. Discuss how HU->Z calibration curves for each planning CT scanner influence the results. It seems calibration curves are not the part of TCIA database. One can expect in a hypothetical setting, that if a patient is CT scanned on two different scanners with two different calibration curves, then the CT images will have different Miu values, but the planned, ground truth dose should be the same. How does this influence the model-predicted dose?

Author Response

Comment 1: Add color bar with units to Figure 11 as it was done for Figure 1.

Response: We have added color bars to Figure 11 as in Figure 1.

Comment 2: Discuss how HU->Z calibration curves for each planning CT scanner influence the results. It seems calibration curves are not the part of TCIA database. One can expect in a hypothetical setting, that if a patient is CT scanned on two different scanners with two different calibration curves, then the CT images will have different Miu values, but the planned, ground truth dose should be the same. How does this influence the model-predicted dose?

Response: As described by the dataset provider, the CT images of the patients were collected from different hospitals. We applied the normalization formula mentioned in the paper to bring the CT images to a common scale (with mean and standard deviation calculated over the entire data). In our previous research at https://www.techscience.com/iasc/v39n2/56497/html, we have discussed in detail the effectiveness of normalizing CT images before feeding them into the model and testing (in section 3.2). Therefore, and also to avoid duplication, we will probably not mention this issue again in the paper.

We have also cited that article to section 2 Data preparation, thanks for the reviewer's comment.

Revised lines: 148-150.
